# Chemical Inhibition of RPA by HAMNO Alters Cell Cycle Dynamics by Impeding DNA Replication and G2-to-M Transition but Has Little Effect on the Radiation-Induced DNA Damage Response

**DOI:** 10.3390/ijms241914941

**Published:** 2023-10-06

**Authors:** Rositsa Dueva, Lisa Marie Krieger, Fanghua Li, Daxian Luo, Huaping Xiao, Martin Stuschke, Eric Metzen, George Iliakis

**Affiliations:** 1Institute of Medical Radiation Biology, University Hospital Essen, University of Duisburg-Essen, 45147 Essen, Germany; lisamariekrieger@yahoo.de (L.M.K.); fanghua.li@uk-essen.de (F.L.); luodaxian@hotmail.com (D.L.); xiao.huaping@mayo.edu (H.X.); 2Institute of Physiology, University Hospital Essen, University of Duisburg-Essen, 45147 Essen, Germany; eric.metzen@uk-essen.de; 3Division of Experimental Radiation Biology, Department of Radiotherapy, University Hospital Essen, University of Duisburg-Essen, 45147 Essen, Germany; martin.stuschke@uk-essen.de; 4West German Proton Therapy Centre Essen (WPE), 45147 Essen, Germany; 5German Cancer Consortium (DKTK), Partner Site University Hospital Essen, 45147 Essen, Germany; 6German Cancer Research Center (DKFZ), 69120 Heidelberg, Germany

**Keywords:** RPA, HAMNO, ionizing radiation, replication, DNA repair, DNA double-strand break

## Abstract

Replication protein A (RPA) is the major single-stranded DNA (ssDNA) binding protein that is essential for DNA replication and processing of DNA double-strand breaks (DSBs) by homology-directed repair pathways. Recently, small molecule inhibitors have been developed targeting the RPA70 subunit and preventing RPA interactions with ssDNA and various DNA repair proteins. The rationale of this development is the potential utility of such compounds as cancer therapeutics, owing to their ability to inhibit DNA replication that sustains tumor growth. Among these compounds, (1Z)-1-[(2-hydroxyanilino) methylidene] naphthalen-2-one (HAMNO) has been more extensively studied and its efficacy against tumor growth was shown to arise from the associated DNA replication stress. Here, we study the effects of HAMNO on cells exposed to ionizing radiation (IR), focusing on the effects on the DNA damage response and the processing of DSBs and explore its potential as a radiosensitizer. We show that HAMNO by itself slows down the progression of cells through the cell cycle by dramatically decreasing DNA synthesis. Notably, HAMNO also attenuates the progression of G2-phase cells into mitosis by a mechanism that remains to be elucidated. Furthermore, HAMNO increases the fraction of chromatin-bound RPA in S-phase but not in G2-phase cells and suppresses DSB repair by homologous recombination. Despite these marked effects on the cell cycle and the DNA damage response, radiosensitization could neither be detected in exponentially growing cultures, nor in cultures enriched in G2-phase cells. Our results complement existing data on RPA inhibitors, specifically HAMNO, and suggest that their antitumor activity by replication stress induction may not extend to radiosensitization. However, it may render cells more vulnerable to other forms of DNA damaging agents through synthetically lethal interactions, which requires further investigation.

## 1. Introduction

Sustained proliferation and replication stress are hallmarks of cancer [1]. Therefore, cancer could be effectively treated by targeting the replication stress response and specifically proteins necessary for DNA synthesis. Replication protein A (RPA) is the major single-stranded DNA (ssDNA) binding protein in eukaryotic cells that is essential for DNA replication and homology-directed DNA repair (HDR) pathways such as homologous recombination (HR) and single-strand annealing (SSA) (reviewed in [2]). DNA end resection (henceforth simply resection) is a vital step of HDR pathways, particularly the HR repair of DNA double-strand breaks (DSB), whereby a long 3′-ssDNA tail is generated by removing nucleotides from the 5′-end. RPA rapidly coats the ssDNA that is generated by the function of helicases during DNA replication or after resection of DSBs, as protection from nucleolytic cleavage and the unproductive folding into secondary structures [3]. Besides an increased structural stability, RPA-coated ssDNA also serves as a checkpoint signaling intermediate [4,5] and is involved in the regulation of the resection rate [6].

RPA is a heterotrimeric protein complex consisting of RPA70/RPA1, RPA32/RPA2, and RPA14/RPA3 subunits with molecular weights of 70 kDa, 32 kDa, and 14 kDa, respectively. RPA70 has a primary role in ssDNA binding and confers protein–protein interactions. A key feature of RPA32 is its N-terminal phosphorylation motif, while RPA14 is thought to provide a structural stability to the complex (reviewed in [2]). RPA becomes phosphorylated during the normal cell cycle by cyclin-dependent kinases (CDKs) [7,8,9] and hyperphosphorylated in response to genotoxic stress by the phosphatidylinositol 3-kinase-related kinase (PIKK) family members ATR, ATM, and DNA-PKcs [2,5,10,11,12,13,14,15,16,17,18]. Therefore, chromatin-bound or hyperphosphorylated RPA is commonly used as a readout of resection [19,20,21,22,23,24,25,26,27].

Considerable activity has been recently directed toward the development of drugs that inhibit the function of RPA with the rationale that they will strongly attenuate cancer cell proliferation, while exerting reduced effects on normal cells owing to their lower proliferation activity. (1Z)-1-[(2-hydroxyanilino)methylidene]naphthalen-2-one (HAMNO) [28], TDRL-505 [29], and fumaropimaric acid (NSC15520) [30,31] are well-characterized small molecule inhibitors targeting RPA70 in vitro, but their potential as cancer therapeutics requires further validation. Moreover, their effect on DNA damage response (DDR) and DSB repair in cells exposed to IR has not been analyzed.

HAMNO selectively binds the N-terminal domain of the largest RPA subunit (70N) [28], potentially inhibiting critical interactions with other proteins that depend on this domain, such as p53, ATRIP, ETAA1, RAD9, MRN, BLM, PrimPol, etc. [32,33,34,35,36,37,38,39,40,41]. Furthermore, the interaction between 70N and phosphorylated 32N [42] is likely abrogated by HAMNO. Glanzer et al. studied the effect of HAMNO on cancer cells and found that HAMNO increased the level of replication stress as detected by elevated pan-nuclear γH2AX staining in S-phase cells. They further showed that HAMNO abrogated the auto-phosphorylation of ATR kinase as well as the phosphorylation of RPA32 at Ser33 by ATR [28].

However, whether HAMNO affects DDR and HDR repair pathways in irradiated cells remains to be investigated. Since HAMNO acts synergistically with etoposide to kill cancer cells [28], we investigate here whether HAMNO functions as a radiosensitizer owing to the anticipated suppressive effects on checkpoint activation and the function of resection-dependent pathways of DSB repair. Our findings extend previous work on the effect of HAMNO on cancer treatment and provide guidelines for the validation of other RPA inhibitors.

## 2. Results

### 2.1. RPA Inhibition by HAMNO Transiently Inhibits Cell Growth and DNA Synthesis

As a first step in our analysis, we measured the growth of A549 cells in the presence of HAMNO (Figure 1A). Untreated cells grew logarithmically. Cells treated with 5 µM HAMNO continued to grow actively with only a small initial delay and eventually reached similar cell numbers in the plateau phase, where cells accumulated in the G1/G0-phase with a consequent reduction in S- and G2/M-phase cells. On the other hand, cells incubated with 20 µM HAMNO showed an initial reduction in doubling time that was overcome after two days and reached markedly lower cell numbers in the plateau phase of growth. The flow cytometry histograms in Figure 1B show that HAMNO had no detectable effects on the cell cycle distribution up to 20 µM, in either phase of growth, i.e., logarithmic or plateau phase. Notably, cells treated with 40 µM HAMNO were unable to divide and began to show signs of apoptosis that became evident as a sub-G1 peak after 3 days (Figure 1A,B). Treatment with 40 µM HAMNO also caused a decrease in the mean cell diameter from ~15 µm to ~11 µm. These results imply that HAMNO can be used for long-term experiments at 5 and 20 µM, while 40 µM causes excessive apoptotic cell death that precludes a mechanistic analysis of its mode of action.

Since RPA is required for replication, we next evaluated the effects of HAMNO on DNA synthesis using EdU incorporation as the endpoint combined with an analysis by flow cytometry. We added 10 µM EdU to the culture medium 20 min before harvesting the cells following different times of exposure to HAMNO (Figure 2A). An analysis of DNA content and EdU fluorescence identified cells in G0/G1-, S-, and G2/M-phases of the cell cycle and allowed the evaluation of the DNA synthesis activity (Figure 2B). While untreated cell cultures contained a fraction of nearly 40% actively replicating S-phase (EdU+) cells that remained relatively constant throughout the experiment, the addition of 5 µM HAMNO transiently decreased the number of EdU+ cells to 5–10% within the first 1 h, which indicated a strong inhibition of DNA synthesis, without affecting the percentage of cells in the S-phase. However, the cells quickly recovered from this replication block and resumed normal levels of DNA synthesis 4 h later (Figure 2C). Upon treatment with 20 µM HAMNO, the percentage of EdU+ cells rapidly dropped to 0% within the first 1 h of treatment and persisted for ~4 h before recovering to control values at 6 h. These data demonstrate that HAMNO causes a concentration-dependent, transient inhibition of DNA replication.

Glanzer et al. showed that HAMNO prevented ATR autophosphorylation and RPA32 phosphorylation following etoposide treatment, thus compromising ATR signaling [28]. We have reported that the inhibition of ATR abrogates IR-induced G2-phase checkpoint activation in cells irradiated in the G2-phase [21,22]. We therefore postulated that HAMNO acts as an ATR inhibitor and designed experiments to test this by probing effects on the G2-checkpoint.

To study activation and recovery of the G2-checkpoint using an exponentially growing cell culture, specifically for cells irradiated in the G2-phase, we employed a two-parameter flow cytometry method combining DNA and H3pS10 staining to detect mitotic cells and quantitate the mitotic index (MI: fraction of cells in mitosis) [21,43,44,45,46]. In unirradiated and untreated cells, the MI remained constant at approximately 2.5% (Figure 3A,B (left)). The addition of 5 µM HAMNO caused a small drop of the normalized MI 2 h after treatment that indicated a small transient inhibition in the progression of G2-phase cells into mitosis that recovered at 6 h. Treatment with 20 µM HAMNO caused a pronounced effect on the MI with the first indications of recovery only at 8 h (Figure 3B (left)).

We next investigated the effect of HAMNO on IR-induced G2-checkpoint activation by exposing replicate cultures to 1 Gy. The results obtained with 5 µM and 20 µM failed to detect any suppression of the G2-checkpoint as it would have been expected from a HAMNO-mediated inhibition of ATR. We conclude that the effects of HAMNO on cell cycle progression and checkpoint regulation derive from mechanisms that are more complex than the mere inhibition of ATR (Figure 3B (right)).

Although the above experiment mainly reflects the response of G2-phase cells, contributions from other phases of the cell cycle cannot be excluded and the response of S-phase cells cannot be analyzed specifically. Therefore, in the next set of experiments, we investigated the effects of HAMNO on IR-induced cell cycle arrest and checkpoint activation in a strictly cell cycle-dependent manner taking advantage of EdU signal. To measure exclusively effects in the S- and G2-phases, EdU was added for 30 min before irradiation (4 Gy) and the fraction of EdU-negative cells (EdU−) was followed as a function of time thereafter (Figure 4A). We observed that HAMNO suppressed, in a concentration-dependent manner, the normally observed IR-induced accumulation of irradiated (4 Gy) cells in the G2-phase (Figure 4B, upper panel).

This effect likely reflects the above-detected, HAMNO-mediated replication inhibition (Figure 2). Indeed, this experiment confirmed that HAMNO abrogated the progression of early and mid S-phase cells into the late S-phase (Figure 4B, lower panel).

For a more in-depth analysis, we carried out kinetics experiments. In the absence of IR, or any other treatment, EdU− cells in the G2-phase divided and depleted the G2 compartment, which manifested as a steady decrease reaching almost zero in the fraction of EdU− G2-cells by 9 h (Figure 4C, upper left panel). The fraction of EdU− G2-cells increased again after 12 h as cells that where in the G1-phase at the start of the experiment (and therefore remained EdU−) arrived in the G2-phase, reaching a new maximum at 15 h. This fraction decreased again when newly arrived cells from the G1-phase also started dividing. The early response of EdU− G2-phase cells remained unaffected by 20 µM HAMNO (Figure 4C, upper left panel), as also did the late response reflecting the cell cycle progression of cells originally in the G1-phase. This result suggests that when G1-phase cells face HAMNO a few hours after treatment inception, their DNA synthesis is not affected, and they reach the G2-phase with kinetics similar to untreated controls. This is perfectly in line with the transient inhibition of DNA replication shown above that indicates recovery even after treatment with a higher HAMNO concentration.

When cells in this experiment were irradiated, no reduction in the EdU− G2-phase fraction was detected up to 10 h, which directly reflects the activation of the G2-checkpoint. At later times, the fraction of EdU− cells in the G2-phase increased, obviously from G1-phase cells reaching the G2-phase. HAMNO had an effect only in this late increase generating a ~5 h delay (Figure 4C, upper right panel). This is again in line with the inability of HAMNO to suppress the checkpoint, despite the anticipated inhibition of ATR.

EdU+ cells in the G2-phase compartment increased as a function of time, as cells from the early and mid S-phase successfully reached the late S-phase, and this increase was delayed by HAMNO (Figure 4C, lower left panel), which is in line with the above documented inhibition of DNA synthesis that generated effects propagating to later times. Radiation exposure of these cells strongly increased the accumulation of HAMNO-untreated cells in the G2-phase as a consequence of the superimposed checkpoint activation. However, this activation remained intact after treatment with HAMNO and was only delayed similarly to the nonirradiated controls, presumably for the same reasons (Figure 4C, lower right panel).

Collectively, these results suggest that HAMNO fails to modify the IR-induced G2-checkpoint both in cells irradiated in the G2-phase as well as in cells irradiated in the S-phase.

### 2.2. HAMNO Enhances IR-Related Resection in S-Phase Irradiated Cells in a Cell Line-Specific Manner

During resection, RPA-coated ssDNA stimulates the activity of ATR. It has been reported that phosphorylated RPA serves as a feedback regulator between DSB end resection and DDR where it impedes resection by inhibiting BLM helicase [6]. Since HAMNO was reported to disrupt RPA phosphorylation [28], we next investigated the effect of the HAMNO treatment on this process. We anticipated that since HAMNO leaves the function of ssDNA-binding domains of RPA70 unaffected, RPA binding to ssDNA may remain sufficiently intact to stimulate ATR activation. On the other hand, RPA unloading or exchange with other proteins such as RAD51 might be affected. To address such potential effects, we used RPA accumulation to chromatin at different times after IR as an endpoint. To discriminate between DSBs arising in the S- and G2-phases, pulse-labeling with EdU was applied 30 min before irradiation as outlined above. Figure 5A depicts the analysis of the IR-induced resection, based on an RPA70 signal analysis with a related antibody. In U2OS cells, 10 Gy of IR caused a substantial increase in chromatin-bound RPA, as compared to nonirradiated cells (Figure 5A).

During the 30 min pretreatment with HAMNO, we detected an increased binding of RPA to chromatin in nonirradiated S-phase A549 cells (Figure 5B, left upper panel)—probably associated with DNA synthesis inhibition (Figure 2), as reported by Glanzer et al. [28]. Moreover, these authors also showed that RPA70 binding to ssDNA was stimulated by HAMNO [28]. In A549 cells, the marked HAMNO-mediated increase in RPA signal in the S-phase (EdU+) was reduced after IR. In contrast to A549 cells, U2OS cells irradiated in the S-phase showed only a modest HAMNO-associated increase in RPA signal that was strongly enhanced after IR (Figure 5B, upper panel). Similar results were obtained with an antibody targeting the middle RPA32 subunit of the RPA complex (Figure 5C).

A549 and U2OS cells irradiated in the G2-phase (EdU−) failed to show HAMNO-induced increases in RPA signal (Figure 5B, lower panel). The observed cell type-specific effect of HAMNO will require further studies. We next assessed the role of RPA inhibition by HAMNO on RAD51 foci formation in response to IR in both A549 and U2OS cells.

### 2.3. HAMNO Abrogates Repair by HR in a Cell Type-Specific Manner

HAMNO binds and blocks the N-terminal domain of RPA70 [28] which is required for RPA protein interactions with multiple proteins involved in DDR and replication stress. During HR repair, RPA is displaced from ssDNA by RAD51 recombinase [47,48,49,50,51,52,53,54,55]. Thus, the increased accumulation of RPA on chromatin in cells irradiated in the S-phase (Figure 5B,C) may be attributed to abrogated RPA interactions with multiple DDR factors. Therefore, we queried the effect of HAMNO on RAD51 foci formation in G2-phase cells using quantitative image-based cytometry (QIBC).

A549 cells analyzed in the S-phase (EdU+) displayed a relatively low (~five) number of RAD51 foci in the absence of IR, which probably reflected collapsed DNA replication forks. This number remained constant during the period of observation (Figure 6A, upper left panel). Notably, after exposure to 4 Gy the number of RAD51 foci strongly increased (~20) at 3 h and subsided only slightly at 6 h. Under these conditions HAMNO reduced the number of RAD51 foci in unirradiated S-phase cells by half, and at 3 h after IR, by nearly 80% (Figure 6A, lower left panel). While HAMNO administration generated similar trends in U2OS cells, the effects failed to reach statistical significance (Figure 6B). The analysis of RAD51 responses in A549 G2-phase (EdU−) cells also indicated similar trends, but there again, the effects were smaller and failed to reach statistical significance (Figure 6A, right panel). Thus, while HAMNO seems to interfere with RAD51 filament formation, the effect is cell line-specific and may derive from excess RPA binding to DNA that is specifically observed in S-phase cells [23,56].

For an alternative investigation of HAMNO’s effects on HR and other DSB repair pathways, we employed GFP reporter cell lines carrying DR-GFP or SA-GFP constructs for monitoring HR and SSA events, respectively. The transient expression of I-SceI endonuclease introduces a DSB, which, when repaired by HR or SSA, creates a functional *GFP* gene. GFP expression is then the readout for the successful repair of these DSBs [57]. Increasing concentrations of HAMNO failed to significantly decrease HR or SSA (Figure 7A). NHEJ was suppressed by about 20% in the EJ5-GFP cell line at increasing concentrations of HAMNO, but the effect again failed to reach statistical significance. The results with the EJ2-GFP cell line, which contains a construct for analyzing alt-EJ events, were variable but showed a tantalizing increase after treatment with HAMNO (Figure 7A). They are interesting vis-à-vis the reported role of RPA in mutagenic microhomology-mediated end joining (MMEJ) by Symington’s lab [58].

Therefore, we next analyzed the contribution of alt-EJ with pulse-field gel electrophoresis (PFGE), which allows the evaluation of DSB induction and repair directly from the generated physical DNA fragmentation. PFGE requires IR doses higher than 5 Gy for fragmentation to be detectable. Consequently, the involvement of HR is undetectable by PFGE since HR operates most efficiently at low DSB loads and is completely suppressed at the high IR doses PFGE requires [23,56]. The DNA-PK inhibitor NU7441 was employed in these experiments to inhibit c-NHEJ and to allow alt-EJ analysis (Figure 7B). Furthermore, since alt-EJ is growth state-dependent [59,60], experiments were performed with cells either in the exponential or the plateau phase of growth. Plateau phase cells are deprived of S- and G2-phase cells, which simplifies their analysis and interpretation, as HR is inactive. In both exponentially growing and plateau phase c-NHEJ-competent cells, DSBs were efficiently rejoined in the presence of HAMNO up to 4 h after IR. In actively growing cells, DSB rejoining after treatment with NU7441 (reflecting mainly alt-EJ) also remained unaffected by HAMNO, while a small suppression that failed to reach statistical significance was observed in plateau phase cells (Figure 7B). In sum, the results obtained suggest that RPA inhibition only marginally inhibits alt-EJ or c-NHEJ; HR appears to be the main DSB repair pathway affected by RPA inhibition, though the effect remains small and is cell line- and cell cycle-dependent.

### 2.4. HAMNO Fails to Radiosensitize A549 Cells

We next examined, by employing clonogenic survival assays, whether HAMNO could radiosensitize tumor cells. When plated immediately after IR and in the presence of HAMNO, at concentrations of 0.1 µM or 0.5 µM, no radiosensitization could be observed. HAMNO at concentrations ≥1.0 µM dramatically reduced the plating efficiency of unirradiated cells. Therefore, experiments using short-term postirradiation exposure to HAMNO were carried out. Cells were pretreated for 1 h with 20 µM HAMNO, irradiated and incubated for 4 or 16 h in the presence of the inhibitor. Cells were then plated for colony formation in cell growth media devoid of HAMNO. The results show that under these conditions, even treatment with 20 µM HAMNO failed to significantly radiosensitize exponentially growing A549 cells (Figure 8A). Since HAMNO exerts its effect mostly in the S-phase of the cell cycle (Figure 2 and Figure 6), we next exposed to IR populations of cells enriched in the S-phase, which is the most radioresistant cell cycle phase [61]. The results showed that even S-phase cells, where HAMNO exerts full effects, failed to be radiosensitized (Figure 8B). We note that in some experiments we observed a slight radioresistance after exposure to HAMNO, which may reflect the effects on the ability of cells to repair potentially lethal damage after stopping DNA replication [62].

## 3. Discussion

RPA is the major eukaryotic single-stranded DNA binding protein and its roles in several aspects of DNA metabolism are well studied. The overexpression of RPA subunits in various cancers is associated with unfavorable treatment outcome. Therefore, the development of RPA protein inhibitors has been pursued intensively and great progress has been made in the last decade (reviewed in [2]). HAMNO is one of the most studied RPA70 inhibitors owing to its promising effects on cancer treatment. Its neutral charge and hydrophobic ring structures enable it to cross the cell membrane more easily than previous RPA70 inhibitors [28].

In this report, we explored the effects of HAMNO on IR-induced DDR. At nontoxic concentrations, the ability of HAMNO to transiently inhibit DNA synthesis resulted in an altered cell cycle progression but did not affect IR-induced G2-checkpoint activation or RPA association with chromatin during the G2-phase. In some experimental settings, we even observed increased RPA binding induced by HAMNO. To our knowledge, it remains unknown whether RPA inhibitors affect RPA’s dynamics on ssDNA, particularly its diffusion along ssDNA and phase separation properties [63,64,65,66].

Increased RPA association with chromatin, i.e., increased resection rate, does not necessarily lead to a proportional increase in Rad51 foci and is not predictive of an increased efficiency of HR. Moreover, the unchanged level of RAD51 foci formation is also not proving an intact HR pathway, especially at radiation doses higher than 4 Gy [23,56]. The increased accumulation of RPA on chromatin may reflect reduced or abolished RPA interactions with multiple DDR factors and particularly its failed replacement by RAD51. Indeed, using QIBC, we showed that in the presence of HAMNO, RAD51 foci formation was significantly reduced in A549 cells but not in U2OS cells, indicating poor RPA exchange with RAD51 and defective repair by HR in a cell line-specific manner.

Experiments using GFP reporter cell lines were unable to confirm that HAMNO has a clear negative impact on the homology-directed DSB repair pathways, HR and SSA. The evidence that RPA inhibition favors alt-EJ requires further studies. Indeed, Deng et al. showed that in budding yeast, RPA prevented the annealing of microhomologies and therefore mutagenic MMEJ—a form of Ku- and ligase IV-independent form of alt-EJ [58]. Nevertheless, HR and SSA are competing mechanisms, and we expected differential results when using the two corresponding reporter cell lines. It should also be noted that the events analyzed with these reporter cell lines occurred over a rather protracted period of time (up to 72 h), which compromised the study of inhibitor effects, particularly when the inhibitor exerts cytotoxicity as HAMNO does—this has actually been reported and analyzed before [67].

Therefore, we employed PFGE to analyze the effects of HAMNO on alt-EJ, which allows the evaluation of DSB induction and repair based on the analysis of actual DNA fragmentation, rather than the indirect assessment of the DSB signaling response, for example by analyzing γH2AX foci formation. The results obtained, however, failed to reveal inhibitory effects either on c-NHEJ or on alt-EJ. However, we speculate that higher concentrations may generate detectable effects.

Mounting evidence suggests that RPA is a critical factor in regulating resection at DSBs. RPA becomes phosphorylated by PIKK family members ATR, ATM, and DNA-PK, where phosphorylation of Ser33 by ATR primes the sequential phosphorylation at other sites [2,5,10,11,12,13,14,15,16,17,18]. The Finkelstein lab reported that phosphorylated RPA suppressed DNA resection [6]. Moreover, RPA phosphorylation is altered by HAMNO; in particular, the etoposide-induced phosphorylation at Ser33 and Ser4/8 is reduced in HAMNO-treated cells [28]. All this led us to suggest that the increased RPA association with chromatin in HAMNO-treated U2OS cells upon irradiation can be attributed to increased resection, which may result in prolonged ssDNA exposure, replication fork collapse, and genomic instability. ATR has been implicated as a key regulator of cell cycle checkpoints. In addition, ATR is the primary kinase that senses replication stress [68,69]. Glanzer et al. reported defective ATR signaling in the presence of HAMNO [28], which in part is in line with our data showing inefficient HR.

Based on these results, we inferred a genomic instability that should potentiate IR-induced cell death. Despite the recently reported HAMNO-induced radiosensitization of glioblastoma cancer stemlike cells [70], we did not observe any significant change in cell survival after irradiation in A549 cells. Nevertheless, we also observed that HAMNO alone impaired cell viability and colony formation as previously reported [28]. The implications of HAMNO as a radiosensitizer are likely cell line-dependent. A promising research line in this direction involves the interaction of RPA with the tumor suppressor p53. The RPA-p53 complex dissociates upon DNA damage induction and is followed by the phosphorylation of both RPA and p53 [71,72]. As p53 is frequently mutated in cancer, it may have an impact on the overall effectiveness of HAMNO as a radiosensitizer.

In conclusion, the current findings extend previous work on HAMNO [28,70,73] by showing that it affects DSB signaling and repair but that these effects are relatively small and cell line- and cell cycle-dependent. Furthermore, at nontoxic concentrations, HAMNO can be used as a transient inhibitor of DNA replication in cell culture experiments. We also confirmed previously reported findings that by itself, HAMNO impaired cell viability and colony formation [28] and further showed that it did not render cells more sensitive to IR. More work will be required to assess the mechanisms by which HAMNO exerts its effects and to develop means to enhance them using agents inhibiting compensating cellular effects with potential to generate synthetic lethality.

## 4. Materials and Methods

### 4.1. Cell Culture and Cell Growth

A549 (human lung adenocarcinoma) and U2OS (human bone osteosarcoma) cells were maintained in McCoy’s 5A medium; U2OS cells harboring DNA repair pathway reporter constructs (DR-GFP, SA-GFP, EJ2-GFP, and EJ5-GFP) [57,74] were kept in Dulbecco’s modified Eagle’s medium (DMEM) supplemented with 10% fetal bovine serum (FBS), 0.1 μg/mL penicillin and 0.1 μg/mL streptomycin at 37 °C in a humidified incubator with 5% CO_2_. All cell lines were routinely tested for mycoplasma contamination and only mycoplasma-free cultures were used in experiments.

For growth curves, cells were plated at a density of 1 × 10^5^ cells per 60 mm dish containing 5 mL of medium and the designated concentration of HAMNO. At the indicated times, two replicates were trypsinized and counted. Cell number was determined by a Multisizer^TM^ 3 particle counter (Beckman Coulter, Krefeld, Germany). The remaining cells were fixed with ice-cold 70% ethanol and used for cell cycle analysis by flow cytometry.

### 4.2. Treatment with IR and Small Molecule Inhibitors

Irradiation of cells was carried out at room temperature (RT) using an X-ray machine (GE-Healthcare, Buckinghamshire, UK) operating at 320 kV and 1.65 mm aluminum filter, at a dose rate of ~3 Gy/min. Once irradiated, the cells were returned to the incubator.

The RPA inhibitor HAMNO (SML1234, Sigma-Aldrich, Merck KGaA, Darmstadt, Germany) was added 30 min before exposure to IR. Control cells were treated with the corresponding concentrations of solvent—mainly dimethyl sulfoxide (DMSO).

For enrichment of cells in the S-phase of the cell cycle, 500 nM CDK4/6 inhibitor PD-0332991 (S1116, Selleck Chemicals, Houston, TX, USA) was applied for 20 h. Four hours after release from PD-0332991, 2 mM thymidine (Sigma-Aldrich, Merck KGaA, Darmstadt, Germany) was added for 16 h. Cells were then released from the block by transferring to fresh medium and were allowed to resume cell cycle progression, which was monitored by FACS analysis.

### 4.3. Flow Cytometry Analysis of DNA Replication by EdU Incorporation

Cells were treated with the indicated concentrations of HAMNO, or left untreated, and pulse-labeled with 10 µM 5-ethynyl-2′-desoxyuridin (EdU), a thymidine analogue, during the last 20 min before harvesting the cells. HAMNO remained in the culture media for the duration of the experiment (Figure 2A). At the indicated times, cells were harvested by trypsinization, spun down, and permeabilized with freshly prepared 0.2% Triton X-100 in phosphate-buffered saline (PBS) for 5 min on ice. Cells were spun down again and fixed with 3% paraformaldehyde (PFA) and 2% sucrose in PBS for 10 min at RT. Samples were stored in PBS containing 0.5% bovine serum albumin (BSA) at 4 °C. Incorporated EdU was conjugated to Cy5 azide by “click” chemistry, and subsequently, cells were stained with a solution of propidium iodide/RNaseA for 15 min at room temperature (RT). Samples were then analyzed on a Gallios flow cytometer (Beckman Coulter, Krefeld, Germany), and raw data were further processed with Kaluza Analysis software v1.3-2.1 (Beckman Coulter, Krefeld, Germany).

### 4.4. Quantification of Mitotic Index by Flow Cytometry

All cell cycle phases, but particularly mitosis, are very sensitive to temperature fluctuations [75]. Therefore, the cell harvesting was carried out with prewarmed solutions at a constant temperature of 37 °C. Since mitotic cells are rounded up and can be easily detached mechanically from the bottom of the culture dish, aspiration was avoided during cell harvesting. Instead, “rounded up” cells were collected by vigorously tapping the culture flask, and growth medium was carefully poured into a 15 mL collection tube. The remaining adherent cells were then harvested with trypsin and transferred to the same collection tube.

For the bivariate flow cytometry (histone H3 phospho-serine 10 (H3pS10) as a marker of mitotic cells and propidium iodide for DNA content), cells were treated with IR and/or small molecule inhibitors as described above and harvested at the indicated time points. Next, cells were fixed and permeabilized in prechilled 70% ethanol overnight at 4 °C. They were then washed with PBS and transferred to blocking buffer (0.5% BSA in PBS) for 1 h at RT. Subsequently, cells were stained with primary antibody against H3pS10 (ab5176, Abcam, Cambridge, UK) diluted 1:1000 in a blocking buffer for 1.5 h at RT, washed with PBS, and stained with secondary antibody coupled with AlexaFluor 488 (Life Technologies, Thermo Fisher Scientific, Waltham, MA, USA) for 1 h at RT in the dark. Finally, cells were washed with PBS, stained in a solution of propidium iodide/RNaseA for 15 min at RT and analyzed using a Gallios flow cytometer (Beckman Coulter, Krefeld, Germany). Raw data were processed with Kaluza software (Beckman Coulter, Krefeld, Germany).

### 4.5. Flow Cytometry Analysis of Chromatin-Bound RPA

Cells were seeded at a density of 0.5 × 10^6^ cells per 100 mm dish for 2 days. Then, 5 µM or 20 µM HAMNO and 10 µM EdU were added 30 min before exposure to IR and subsequently, the EdU-containing media were replaced with fresh prewarmed cell culture media, devoid of EdU but containing or not HAMNO, for the remainder of the experiment. At different times after IR, cells were harvested by trypsinization and centrifuged for 5 min at 1200 rpm. Unbound RPA was extracted by resuspending the cell pellet in a freshly prepared permeabilization solution (0.2% Triton X-100 in PBS) for 5 min on ice, followed by a spin-down at 1800 rpm for 5 min. The cell pellet was resuspended in a fixation solution containing 3% PFA plus 2% sucrose and incubated for 15 min at RT. Cells were centrifuged and blocked with PBS containing 0.5% BSA overnight at 4 °C. Thereafter, cells were stained under gentle agitation with primary anti-RPA70 or anti-RPA32 mouse monoclonal antibody obtained by hybridoma technology [76] for 1.5 h, and subsequently with a secondary antibody conjugated with Alexa Fluor 488 for 1 h, both diluted in a blocking buffer. The incorporated EdU was conjugated to Cy5 azide by “click” chemistry. Subsequently, the DNA was stained by incubating cells with a solution of propidium iodide/RNaseA for 15 min at RT. Samples were analyzed using a Gallios flow cytometer (Beckman Coulter, Krefeld, Germany).

### 4.6. I-SceI-Based Reporter Assays for Assessing DSB Repair by Different Repair Pathways

The following U2OS-GFP reporter cell lines that monitor the processing of I-SceI-induced DSBs by a distinct repair pathway were used: DR-GFP cell line measured the efficiency of HR, EJ5-GFP reporter measured c-NHEJ, and EJ2-GFP and the SA-GFP cell lines were used to monitor alt-EJ and SSA, respectively [57,74].

Cells were harvested and centrifuged at 1200 rpm for 5 min. In total, 1 × 10^6^ cells were nucleofected with 1 µg I-SceI-expressing plasmid using Amaxa’s Nucleofector (Lonza Bioscience, Basel, Switzerland). Subsequently, cells were seeded in prewarmed medium either containing or devoid of HAMNO. After 24 h, cells were collected with trypsin, and the GFP signal arising from repair events was quantitated using a Gallios flow cytometer (Beckman Coulter, Krefeld, Germany). The frequency of DSB repair in HAMNO-treated cells was calculated relative to the corresponding value for DMSO-treated cells.

### 4.7. Immunofluorescence and Quantitative Image-Based Cytometry (QIBC)

Cells were grown on coverslips and seeded at a density of 0.15 × 10^6^ cells per 30 mm dish for 2 days. Then, 20 µM HAMNO and 10 µM EdU were added 30 min prior to IR exposure. Subsequently, EdU-containing cell culture media were replaced with fresh prewarmed EdU-free media but containing or not HAMNO, for the remainder of the experiment. At different times after IR, cells were fixed with 2% PFA in PBS for 15 min, washed once with PBS, and then incubated in a freshly prepared prechilled permeabilization solution (100 mM Tris pH 7.4, 50 mM EDTA pH 8.0 and 0.5% Triton X-100) for 7 min at RT. Subsequently, cells were washed again with PBS and blocked in PBS containing 0.5% BSA overnight at 4 °C. Cells were then stained with primary anti-Rad51 antibody (GeneTex, Irvine, CA, USA) or anti-RPA32 [76] for 1.5 h and with Alexa Fluor^®^ 647-conjugated secondary antibody (Life Technologies, Thermo Fisher Scientific, Waltham, MA, USA) for 1 h, both diluted in a blocking buffer. The incorporated EdU was conjugated to 6-FAM azide by “click” chemistry. Subsequently, the DNA was stained by incubating cells with PBS containing 200 ng/mL 4′,6-diamidino-2-phenylindole (DAPI) for 20 min. Finally, cells were washed with PBS and the coverslips mounted with the ProLong Gold antifade mountant (Life Technologies, Thermo Fisher Scientific, Waltham, MA, USA) on microscope slides. For the cell cycle-dependent analysis of repair foci formation by QIBC, samples were scanned on an AxioScan.Z1 slide scanner (Zeiss, Oberkochen, Germany) as described before [23]. The image analysis was performed using Imaris 9.5.1 software (Bitplane, Belfast, UK), and the data generated were processed by open-source Orange software v3.3 (University of Ljubljana, Ljubljana, Slovenia).

### 4.8. Clonogenic Survival Assay

In all experiments, exponentially growing A549 cells were treated with the indicated concentrations of HAMNO for 30 min and exposed to different doses of IR. Cells were then seeded for colony formation immediately after irradiation in the presence of the inhibitor. For delayed plating (DP), cells were allowed to repair DNA damage for 4 h or 16 h in the presence of HAMNO and then plated for colony formation in HAMNO-free media. Colonies were stained with 1% crystal violet dissolved in 70% ethanol and counted under a stereomicroscope. The plating efficiency (PE) of untreated cells was calculated as the ratio of the number of colonies counted to the number of cells seeded. The surviving fraction (SF) at a given dose was calculated as the ratio of the number of colonies that arise after the treatment to the number of cells seeded, normalized to the PE of unirradiated controls.

### 4.9. Pulse-Field Gel Electrophoresis (PFGE)

PFGE was used to assess the induction and repair of DSBs based on DNA fragmentation as previously described [77,78]. Ethidium bromide (EtBr)-stained agarose gels were scanned with a Typhoon Imager (GE Healthcare, Buckinghamshire, UK) and the fraction of DNA released (FDR) from the plug into the lane was quantified by ImageQuant 5.2 software (GE Healthcare, Buckinghamshire, UK). Dose–response curves (FDR value versus IR dose) were used to estimate dose-equivalent (Deq) values for each FDR measured at a given time point. Repair kinetics are presented as Deq versus repair time.

### 4.10. Statistical Analysis

SigmaPlot software v12.5 was used for all analyses and the different groups were compared for statistically significant differences employing Student’s *t*-test, or a one-way ANOVA, followed by Dunn’s or Holm–Sidak’s method for multiple comparisons. A nonparametric Mann–Whitney *U* test or Kruskal–Wallis one-way ANOVA on ranks was applied where appropriate. *P* < 0.05 was considered to be significant, and the *p*-values are provided as follows: * *p* < 0.05, ** *p* < 0.01, *** *p* < 0.001, ns (not significant).

## Figures and Tables

**Figure 1 ijms-24-14941-f001:**
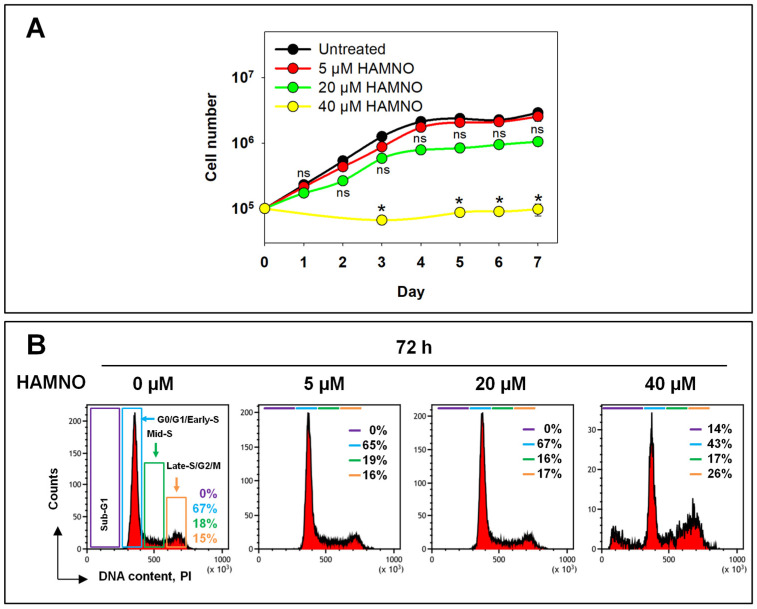
Cell growth of A549 in the presence of the RPA inhibitor HAMNO. (**A**) Growth curves of cells in the presence of increasing concentrations of HAMNO presented as cell number per dish as a function of time. The data were statistically analyzed by a Kruskal–Wallis one-way ANOVA on ranks and Dunn’s post hoc test within each time point, * *p* < 0.05, ns (not significant). (**B**) Representative cell cycle histograms of untreated and HAMNO-treated cells after three days of treatment. DNA was stained with propidium iodide (PI).

**Figure 2 ijms-24-14941-f002:**
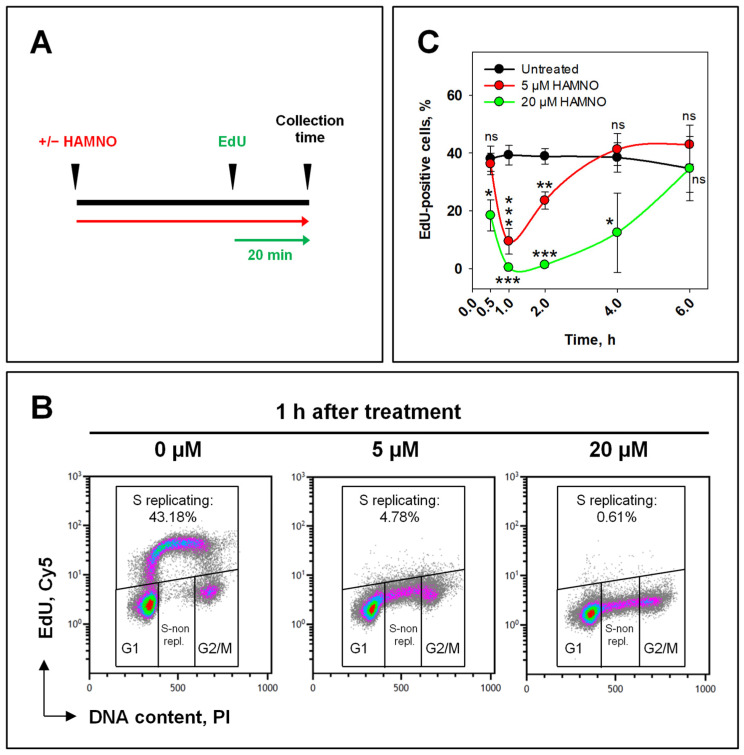
RPA inhibition by HAMNO halts DNA synthesis. (**A**) Schematic representation of EdU labeling of S-phase cells. A549 cells were treated or not with HAMNO (red) and subsequently pulse-labeled with 10 µM EdU (green) during the last 20 min in the presence of HAMNO before collecting the cells. (**B**) Representative two-parameter density plots of PI vs. EdU−Cy5 fluorescence. The percentage of replicating S-phase cells (EdU+) is indicated, which reflects the levels of DNA synthesis. Unimpeded (left) or impeded (middle and right) DNA synthesis results are shown. (**C**) Quantification of DNA synthesis (EdU+) over time. Error bars indicate the standard deviation of three independent experiments. Statistical significance was determined by a one-way ANOVA within each time point, followed by Holm–Sidak’s post hoc test (* *p* < 0.05, ** *p* < 0.01, *** *p* < 0.001, ns (not significant)).

**Figure 3 ijms-24-14941-f003:**
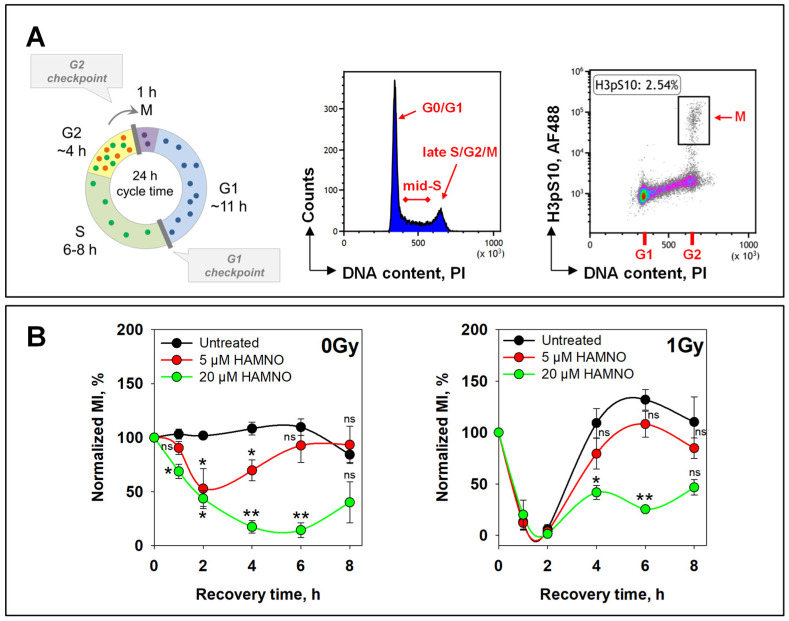
RPA inhibitor HAMNO attenuates the progression of G2-phase cells into mitosis. (**A**) Left: The eukaryotic cell cycle. The duration of the phases refers to human cells. The G2-checkpoint can be subdivided into that activated in cells irradiated in the G2-phase and that activated in cells irradiated in the S-phase. Middle: Cell cycle distribution of A549 cells determined by the intensity of propidium iodide (PI) staining. Right: G2-checkpoint activated specifically in cells irradiated in the G2-phase can be studied by combined DNA (x axis) and phosphorylated histone H3 at Ser10 (y axis) staining that allows the discrimination between G2 and mitotic (M) cells for up to 6 h after IR; the percentage of cells in mitosis is indicated. (**B**) Exponentially growing A549 cells were pretreated with HAMNO 30 min before exposure to IR and harvested at different times post-IR. The mitotic index (MI) was determined as the percentage of cells in mitosis against the total number of cells. Statistical significance was determined by a one-way ANOVA within each time point, followed by Holm–Sidak’s post hoc test (* *p* < 0.05, ** *p* < 0.01, ns (not significant)).

**Figure 4 ijms-24-14941-f004:**
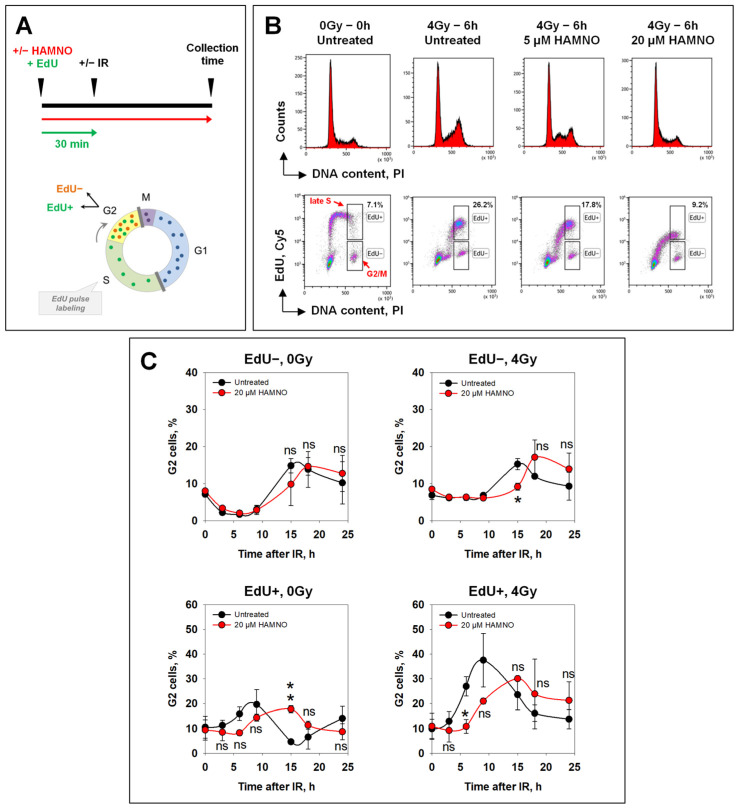
RPA inhibitor HAMNO slows down progression through the cell cycle. (**A**) Schematic representation of EdU labeling. Exponentially growing A549 cells were pulse-labeled with 10 µM EdU 30 min before exposure to IR in the presence or not of HAMNO. After irradiation, EdU-containing medium was exchanged with fresh prewarmed medium supplemented or not with HAMNO. (**B**) Upper panel: Cell cycle profiles of A549 cells pretreated with the indicated concentrations of HAMNO and irradiated or not with 4 Gy. Lower panel: Flow cytometry density plots of cells pulse-labeled with EdU for 30 min before irradiation. (**C**) Cells were harvested at different times post IR. Percentage of EdU+ (upper panel) or EdU− (lower panel) G2-cells is plotted against time. Statistical significance was determined by Student’s *t*-test within each time point (* *p* < 0.05, ** *p* < 0.01, ns (not significant)).

**Figure 5 ijms-24-14941-f005:**
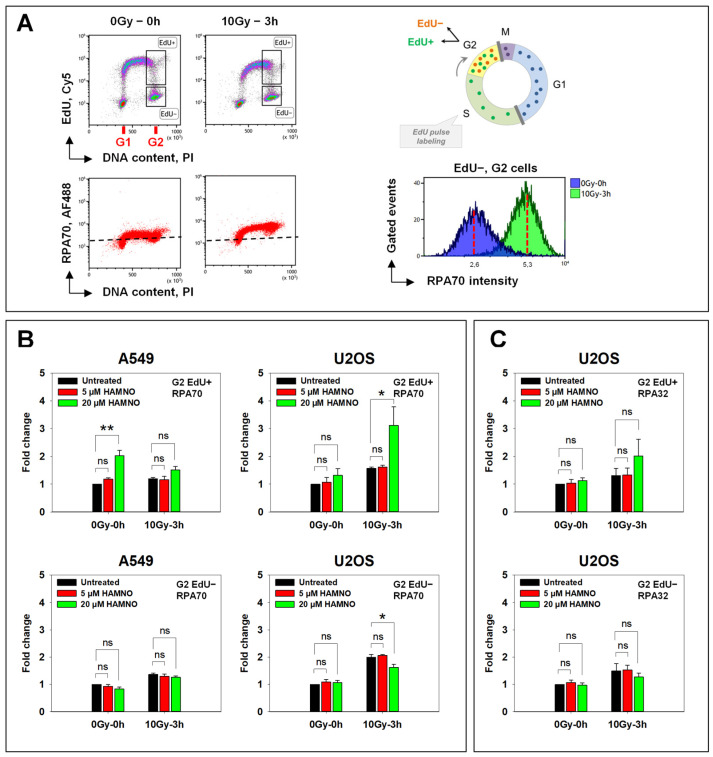
Impact of RPA inhibitor HAMNO on RPA association with chromatin. (**A**) Left: Flow cytometry density plots of U2OS cells based on DNA dye PI, EdU labeled with far-red fluorescent Cy5 azide via the Click-iT reaction and chromatin-bound RPA70 labeled with primary monoclonal anti-RPA70 antibody followed by Alexa Fluor 488-conjugated secondary antibody. Gates are set to define cell populations of interest. Right: Representative histogram showing RPA70 signal in nonirradiated versus irradiated G2-phase cells. (**B**) RPA70 signal intensity in A549 and U2OS cells irradiated in the S- (EdU+) or G2-phase (EdU−). (**C**) RPA32 signal intensity in U2OS cells sustaining DNA damage in the S- (EdU+) or G2-phase (EdU−). RPA intensities in (**B**,**C**) are represented as fold change with respect to the untreated nonirradiated control at 0 h. Data are compiled from at least two independent experiments (±SD). Statistical significance was determined by one-way ANOVA and Holm–Sidak post hoc tests (* *p* < 0.05, ** *p* < 0.01, ns (not significant)).

**Figure 6 ijms-24-14941-f006:**
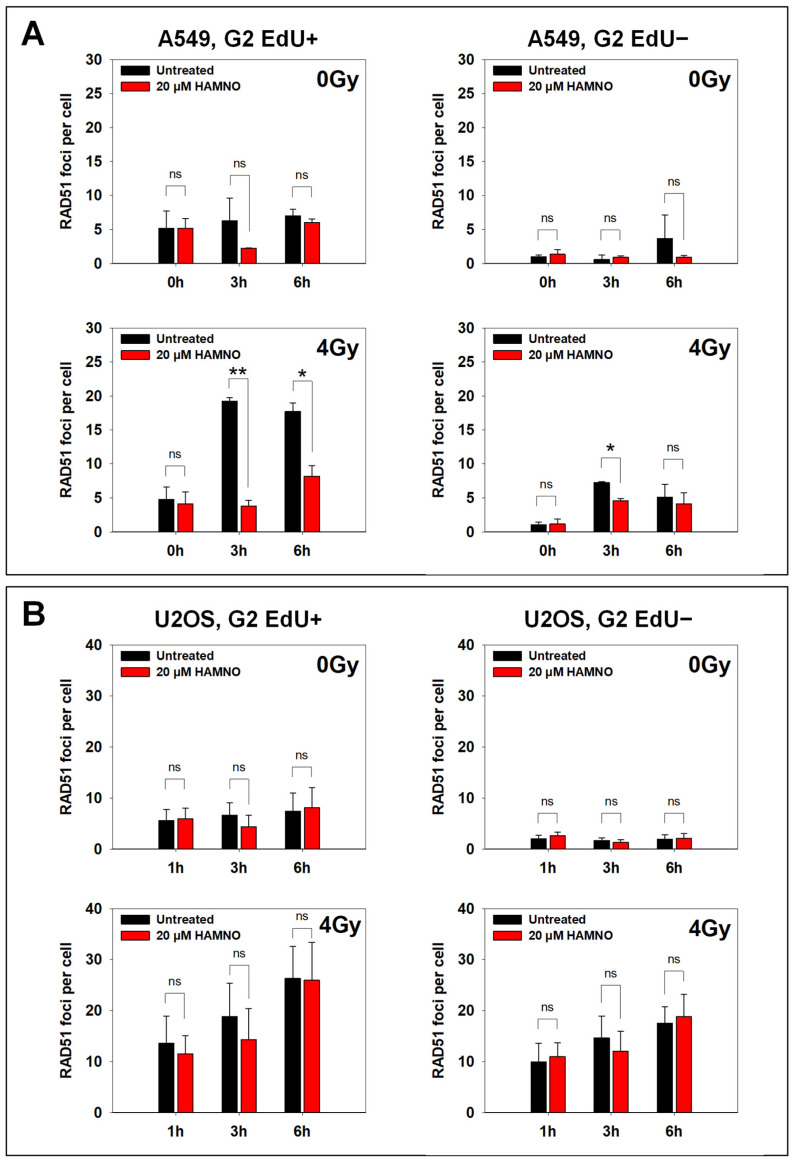
RPA inhibitor HAMNO abrogates repair by HR in a cell line-specific manner. A549 (**A**) and U2OS (**B**) cells were pulse-labeled with EdU 30 min before exposure to IR in the presence or absence of HAMNO. The EdU-containing medium was then exchanged with fresh prewarmed medium supplemented or not with HAMNO. Cells were fixed at the indicated time post-IR, stained with anti-RAD51 antibody, Click-iT EdU, and DAPI and analyzed by QIBC. The data were statistically analyzed by Student’s *t*-tests, * *p* < 0.05, ** *p* < 0.01, ns (not significant).

**Figure 7 ijms-24-14941-f007:**
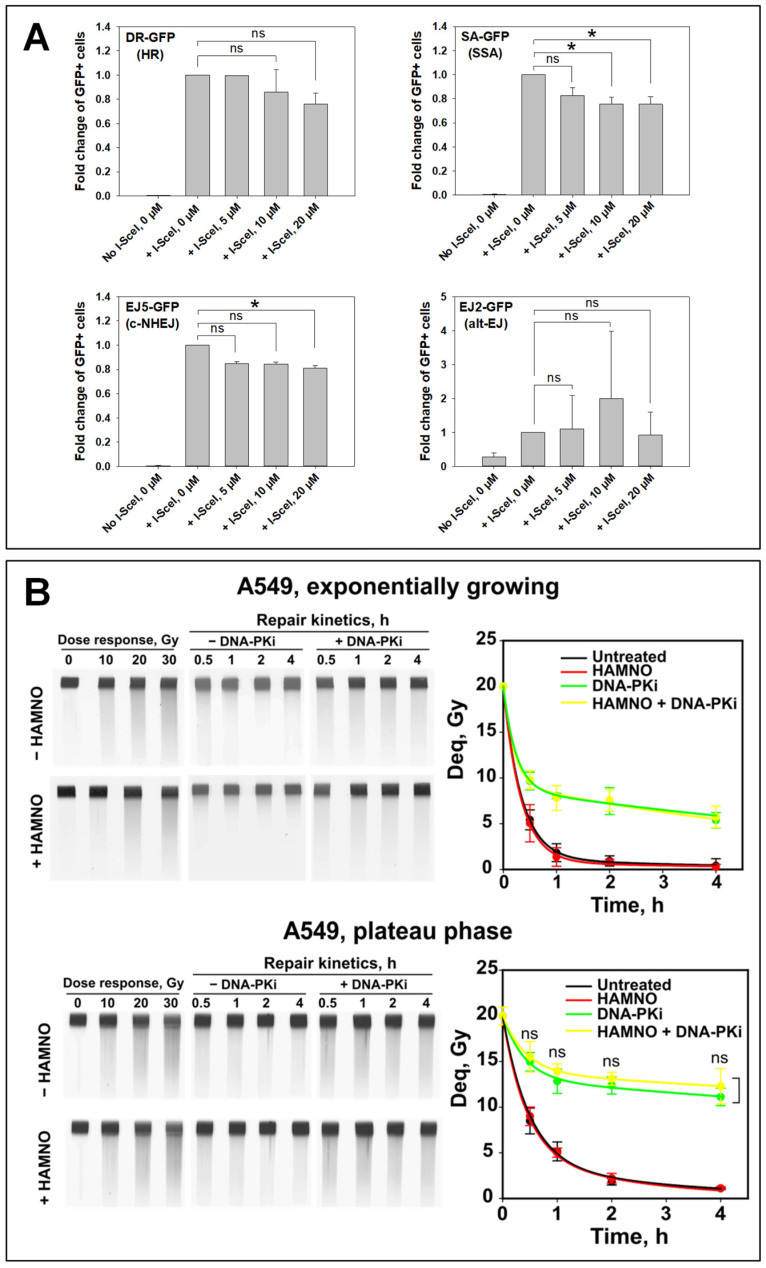
Impact of RPAi HAMNO on DSB rejoining by distinct repair pathways. (**A**) U2OS reporter cells used to monitor HR (DR-GFP), SSA (SA-GFP), c-NHEJ (EJ5), and alt-EJ (EJ2) events. GFP-positive cells were measured by flow cytometry 24 h after Nucleofection^®^ with an I-SceI expression plasmid. Immediately after Nucleofection^®^, cells were transferred to media containing different concentrations of the RPA inhibitor HAMNO, which remained in the media until analysis. The percentage of GFP-positive cells is plotted as fold change with respect to the untreated cells (0 µM) expressing I-SceI. The data were analyzed by a Kruskal–Wallis one-way ANOVA on ranks and Dunn’s post hoc test, * *p* < 0.05, ns (not significant). (**B**) Analysis of DSB rejoining using PFGE. The DNA-PKcs inhibitor NU7441 was applied to compromise c-NHEJ and allow the analysis of alt-EJ. ns (not significant) as determined by Student’s *t*-test. Representative PFGE gels are shown on the left.

**Figure 8 ijms-24-14941-f008:**
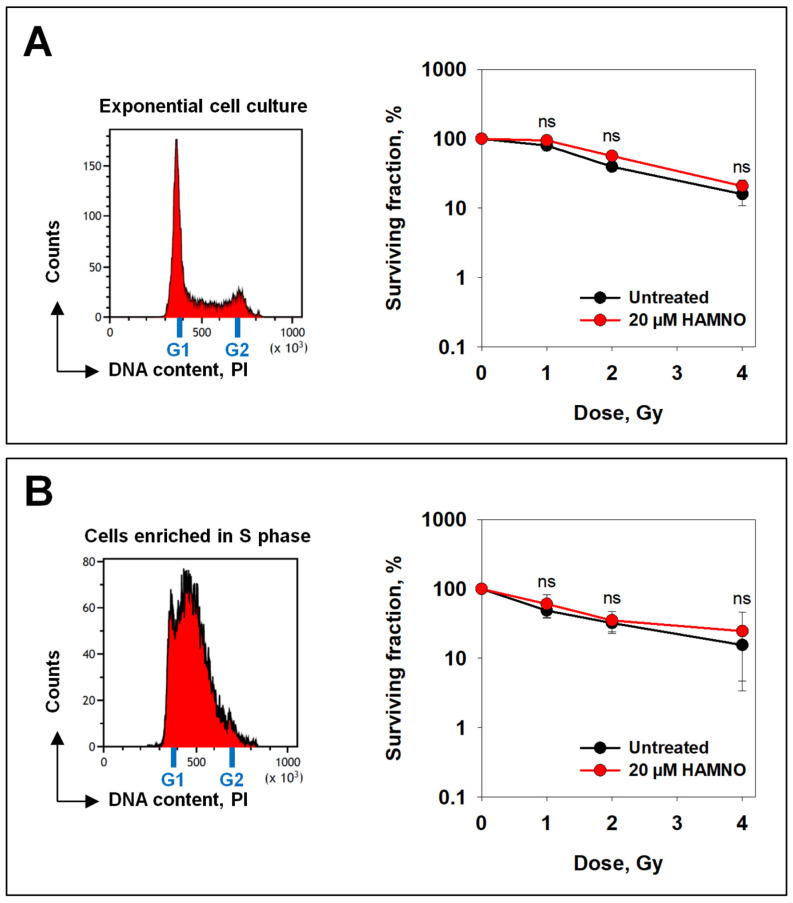
HAMNO fails to radiosensitize A549 cells. (**A**) Exponentially growing cells were treated with HAMNO 30 min before irradiation. Cells were then allowed to repair for 16 h and plated in the absence of HAMNO. (**B**) Cells were enriched in the S-phase by treating them with the CDK4/6 selective inhibitor PD-0332991 followed by a single thymidine block. After release in fresh medium, cells were exposed to IR, allowed to repair for 4 h and plated in the absence of HAMNO. ns (not significant) as determined by Mann–Whitney *U* tests.

## Data Availability

The data presented in this study are available on request from the corresponding author.

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
