# Peer review of "Chemical Inhibition of RPA by HAMNO Alters Cell Cycle Dynamics by Impeding DNA Replication and G2-to-M Transition but Has Little Effect on the Radiation-Induced DNA Damage Response"

_ijms, 2023, doi:10.3390/ijms241914941_

Round 1

Reviewer 1 Report

An article "Chemical inhibition of RPA by HAMNO alters cell cycle dynamics by impeding DNA replication and G2-to-M transition but has little effect on the radiation-induced DNA damage response" by Dueva et al. presents data on the effects of HAMNO on cells exposed to ionizing radiation (IR), focusing on effects on DNA damage response and the processing of DSBs and explore its potential as a radiosensitizer. The study is performed very well, data are technically sound. The Authors performed a sophisticated analysis of the cell cycle progression. The results are meticuluously analyzed and properly presented. The conclusions are supported by the results, and appropriately discussed, although minor revision of the discussion is necessary..

Minor comment:

1. Please unify the labels in the figures e.g., "5 uM" or "5uM" can be found (Figure 5). 

2. Repetition of the results (with annotations to specific figures) in the discusssion should be removed, and can be replaced by extended reference to cited literature.

Reviewer 2 Report

The manuscript aimed to examine the impact of HAMN), the RPA70 inhibitor, on cell cycle distribution, DNA synthesis, and radiosensitization of A549 lung cancer cells in vitro. 

I have the following suggestions and concerns regarding this manuscript:

1) FACs data shown in  Figure 1B is not convincing, since non-treated cells over 6 day culture period demonstrated a cell cycle profile similar to the cancer cells treated with 5 and 20 microM of HANMO. Moreover, an increased number of cells in G0/G1 phases over a 6-day culture period might result from the 100% confluency of the monolayers reached over this period. Thus, the figure did not provide any useful info. 

2) Since the number of mitotic (i.e. pH3Ser10-positive) cells in control is very low (2.5%), it's difficult to make a strong conclusion about the effect of HAMNO on IR-induced G2-checkpoint activation 

3) The major weakness of the manuscript (which was also mentioned in the title, abstract, and Results section, as well) is the low effect of RPA inhibitor to rediosensitize cancer cells.  The authors have to provide an explanation of this, especially after providing the data based on the functional assays aimed to examine the efficacy of homology-mediated DNA repair by using the corresponding reporter cell lines.

4)  The original immunofluorescence images illustrating the focal distribution and co-localization of the DNA repair proteins (e.g. H2AXSer139, Rad51, RPA 70, RPA32 (Ser4/8), the components of MRN complex (i.e. Mre11, Nbs1) are highly desirable to convince the effects of HAMNO. The decreased intensity of the signal measured by FACs might be not accurate. 

5)  The manuscript also lacks the original blots illustrating the expression of activated (phosphorylated) and total forms of DNA repair proteins involved in the repair of DNA DSBs induced by radiation.   

6) The study was performed by using one cancer cell line only.

7) The experiments with siRNA RPA70 might be useful as the positive control for radiosensitization data. 

Minor:

Typos:  "mere" (line 177), RPA2 (line 143), etc. 

Round 2

Reviewer 2 Report

The authors responded to the comments and suggestions. Even some of the data is still missing in the manuscript, the authors provided the references to fill some of these gaps and discussed the potential DDR mechanisms shown in the manuscript. 

The quality of the manuscript was improved and it's suitable for publication in present form.